# Continuous Gait Phase Estimation for Multi-Locomotion Tasks Using Ground Reaction Force Data

**DOI:** 10.3390/s24196318

**Published:** 2024-09-29

**Authors:** Ji Su Park, Choong Hyun Kim

**Affiliations:** 1Safety Component R&D Center, Gyeonggi Regional Division, Korea Automotive Technology Institute, Siheung-si 15014, Republic of Korea; indicepark@katech.re.kr; 2Center for Bionics, Korea Institute of Science and Technology, Seoul 02792, Republic of Korea

**Keywords:** continuous gait phase estimation, ground reaction force, force sensing resistors, bidirectional long short-term memory, insole device, gait analysis

## Abstract

Existing studies on gait phase estimation generally involve walking experiments using inertial measurement units under limited walking conditions (WCs). In this study, a gait phase estimation algorithm is proposed that uses data from force sensing resistors (FSRs) and a Bi-LSTM model. The proposed algorithm estimates gait phases in real time under various WCs, e.g., walking on paved/unpaved roads, ascending and descending stairs, and ascending or descending on ramps. The performance of the proposed algorithm is evaluated by performing walking experiments on ten healthy adult participants. An average gait estimation accuracy exceeding 90% is observed with a small error (root mean square error = 0.794, R^2^ score = 0.906) across various WCs. These results demonstrate the wide applicability of the proposed gait phase estimation algorithm using various insole devices, e.g., in walking aid control, gait disturbance diagnosis in daily life, and motor ability analysis.

## 1. Introduction

Gait analysis involves the analysis of human body movements while walking, and gait phase estimation is the most basic gait analysis technique. Continuous gait phase estimation (cGPE) is usually used to estimate gait phases continuously during gait disturbance diagnosis, walking rehabilitation [1,2,3,4,5,6,7], and the assessment of control over exoskeleton robots or walking aids [8,9,10,11,12].

The commonly used cGPE methods can be categorized into three main types. The first comprises estimation methods that use the linear relationship between joint angle and/or angular velocity information and the gait phase [8,9,13,14,15]. These methods usually estimate gait phases by attaching inertial measurement units (IMUs) to the femoral region or the calf of subjects. They are generally utilized for level-walking analysis; their application to multilocomotion tasks (e.g., stairs and ramps) is difficult [15,16].

Methods of the second type estimate gait phases by analyzing the change cycle of biosignal information measured during walking using an adaptive oscillator (AO) [17,18,19,20,21,22]. AO is a non-model-based method that expresses the cGPE result as a gait phase value in the range [0, 1]. However, AO-based cGPE cannot adapt to changes in walking speed; thus, the estimated value may diverge [19].

Methods of the third type address the shortcomings of the first two types. They are used to estimate gait phases using machine learning [12,23,24,25,26,27,28,29,30]. For example, Lee et al. proposed a Bi-LSTM model that estimates gait phases using data measured through IMUs mounted on the femoral region and waist of subjects [23]. Lu et al. estimated gait phases using long short-term memory (LSTM) corresponding to varying ground slopes; however, they did not consider the boundary conditions of level ground and ramps [30]. Choi et al. proposed a neural network domain for estimating gait phases using an unsupervised learning model that does not rely on gold-standard gait phase data [27].

Existing machine learning-based cGPE studies exhibit two characteristics. First, they are commonly used in connection with controlling exoskeleton robots or prosthetic legs. In this case, cGPE results are obtained based on joint angle and/or angular velocity information acquired using an IMU attached to artificial joints [13,14,15,16,26,27,29,30]. Therefore, most research on controlling auxiliary devices has used cGPE algorithms based on IMUs [17,18,19,23,25,28].

Although other sensors have also been used, they are mostly minor sensors that are utilized as part of composite sensors or used to provide learning reference values [20,22]. Few studies have obtained cGPE results by focusing on biosignals measured using everyday footwear [20,31].

Second, only simple walking environments are considered in experiments in existing machine learning-based cGPE research. In other words, changes in walking speed or walking conditions (WCs) have not been considered adequately [15,22,23,30]. In some studies, walking experiments were performed under two WCs (level ground and stairs) using IMUs [19,28]; however, there are only a few such studies.

In this study, we developed a cGPE algorithm using force sensing resistors (FSRs) data while also classifying various WCs. High cGPE accuracy is secured under varying WCs (multilocomotion tasks) by proposing a cGPE algorithm that analyzes FSR data using an LSTM. The proposed algorithm consists of two main stages: WC classification during walking experiments and cGPE based on the classified WCs. The performance of the proposed algorithm is verified using walking experiments involving ten healthy adult subjects.

The contributions of this study to technology development in cGPE research can be summarized as follows:Real-time classification of WCs is performed by analyzing FSR data using machine learning methods.Accurate real-time cGPE is performed to correspond to continuously varying WCs using a combination of WC classification results and FSR data.Walking experiments involving healthy adults are performed to verify the source-specific application of the proposed cGPE, irrespective of WCs.

## 2. Methods

In this section, a data acquisition system (DAS) for FSR data is introduced, and the experimental protocol, subject information, and cGPE algorithm are described.

### 2.1. DAS

Figure 1a depicts the insole device, comprising an insole containing the FSRs (FSR 402, Interlink Electronics, Inc., Irvine, CA 93012, USA) used to acquire FSR data, a printed circuit board (PCB) used to collect and store FSR data, and the shoes containing the insole device. The insole device is designed to transmit the measured FSR data wirelessly from the left foot to the right foot, synchronize the data, and wirelessly transmit the entire FSR dataset to a personal computer (PC). The microprocessor unit (MPU) mounted on the PCB is an STM32F411x (STMicroelectronics, Marlow, UK), with a data sampling rate of 100 Hz. The FSRs are installed at the toe, first metatarsal bone, 5-th metatarsal bone, cuboid bone, and heel, as in a previous study [32]. In aggregate, ten units are used on each foot.

### 2.2. Experimental Conditions and Protocol

In this study, five WCs are considered, as depicted in Figure 1b: level walk (LW), stair ascent (SA), stair descent (SD), ramp ascent (RA), and ramp descent (RD).

Three experimental walking scenarios are created by combining various WCs, as depicted in Figure 1c. While performing the walking experiments, the operator walks behind the subject with a laptop to collect the experimental data. Changes in WC are tallied by pressing the space bar every time they happen, and this signal is stored alongside the FSR data for subsequent gait analysis.

Each experiment is performed three times for scenario 1, three times in the forward direction, and three times in the reverse direction for scenarios 2 and 3.

Ten healthy adults are included in the walking experiments: five males (age 25 ± 2 years, height 176.0 ± 4.5 cm, and weight 74.3 ± 9.7 kg, mean ± standard deviation (STD)) and five females (age 21 ± 6 years, height 165.6 ± 9.8 cm, and weight 62.3 ± 11.7 kg). In the experiment, each subject was asked to participate in the three aforementioned scenarios at a walking speed that they were comfortable with.

The experimental protocol was approved by the Institutional Review Board (IRB) of the Korea Institute of Science and Technology (approval no. KIST-202309-HR-001). Moreover, all participants provided written informed consent for this study prior to participation.

### 2.3. Walking Condition Classification and cGPE

Bi-LSTM is used in the proposed algorithm, and the WC classification model and the cGPE algorithm are connected in series, as depicted in Figure 2.

In the WC classification model, ten FSR sensor data points are received as input values, and one of the five WCs (LW, SA, SD, RA, and RD) is classified and exported as the output value. Two ground surfaces (paved and unpaved roads) are mixed in the RA/RD conditions; however, they are not classified or learned.

The proposed cGPE algorithm estimates gait phases based on four types of data: the WC classification results exported from the WC classification model, the individual FSR data obtained from the ten FSRs, the sum of the FSR data obtained from three FSRs (toe, first metatarsal bone, and 5th metatarsal bone) in the front part of the insole, and that corresponding to two FSRs (cuboid bone and heel) in the back part (FSR_Fore_ and FSR_Back_, respectively) of the insole.

Here, FSR_Fore_ and FSR_Back_ are calculated using Equations (1) and (2) and normalized by dividing them by FSR_Fore.max_ and FSR_Back.max_.
FSR_Fore_ = FSR_Toe_ + FSR1st_-meta_ + FSR5th_-meta_(1)
and
FSR_Back_ = FSR_Cub_ + FSR_Heel_.(2)

The starting point of the gait phase is defined as the point at which the FSR data from the right-foot FSR_Heel_ increases. Thus, the gait phase can be expressed using a value between 0% and 100%. When this value is substituted into Equations (3)–(5), the continuous sinusoidal function (CSF) value is obtained. This value is used as input data for machine learning models [23,28].
(3)θ=Gaitphase×(2π/100),
(4)Output1=cos⁡θ,
and
(5)Output2=sin⁡θ.

Table 1 lists the parameters of the Bi-LSTM network architecture used for cGPE. The Adam optimizer is employed with a learning rate of 0.001 for training. The batch size is 100. A sequence length of 100 is used for the WC classification model and 25 for the cGPE model. Categorical cross-entropy is used as the loss function for the Walking Condition Classification model, and mean squared error (MSE) is used for the cGPE model.

Among the FSR data, the third data point of each scenario is used for testing, while 80% and 20% of the remaining data are used for training and validation, respectively.

## 3. Results

### 3.1. Experimental Measurement of FSRs

FSR data are normalized using the max–min method. They are then grouped by sequence size for learning, and no noise filters are applied. Figure 3 depicts the FSR data of subject #1, which are subjected to linear interpolation based on each stride and then expressed with respect to varying WCs. The use of DAS, which is adopted in this study, makes it possible to secure FSR data based on walking experiments.

### 3.2. Walking Condition Classification Results

In the first Bi-LSTM model for WC classification, data learning for the WCs is terminated when no improvement is observed in validation loss over five epochs during five-fold cross-validation.

Table 2 lists the accuracies of the WC classification models described in Table 1. All estimation results obtained from all ten subjects in the walking experiments are presented. The WC classification accuracy based on FSR data is 90 ± 1%, 91 ± 4%, 90 ± 5%, 90 ± 3%, and 91 ± 3% for LW, SA, SD, RA, and RD, respectively, confirming the distribution of true positive data.

### 3.3. cGPE Results

The performance of the proposed cGPE algorithm is evaluated in terms of root mean square error (RMSE) and R^2^ score between the estimated and actual cGPE values. An R^2^ score close to 1 indicates high-quality estimation performance.

The cGPE algorithm is trained using FSR data acquired from all subjects. However, both own-subject and cross-subject tests are conducted to validate model training. A cross-subject test is conducted using the data obtained from the third experiment among the FSR data obtained under the three walking experiment scenarios for each subject.

The performance of the cGPE algorithm is described in Table 3 and Table 4, which classify the subjects into two groups: Subjects 1–5 and Subjects 6–10.

The RMSE for the own-subject test is observed to be 0.794 (the RMSE values of LW, SA, SD, RA, and RD are 0.658, 0.939, 0.917, 0.682, and 0.775, respectively), whereas that for the cross-subject test is 2.434 (with corresponding RMSEs of 1.845, 3.365, 2.842, 2.144, and 1.974 respectively), which is 67% higher than that of the own-subject test.

The R^2^ score of the estimated cGPE is 0.906 (0.922, 0.889, 0.892, 0.920, and 0.908, respectively) in the own-subject test, and 0.728 (0.780, 0.612, 0.723, 0.758, and 0.767, respectively) in the cross-subject test. In other words, practically applicable gait phase estimation performance is observed in the own-subject test, as the average R^2^ score exceeds 0.9, but the cGPE performance is evaluated to be relatively poor in the cross-subject test.

The applicability of the cGPE algorithm only to its own subjects is also confirmed in Figure 4. In Figure 4, the CSF value is converted into a phase variable using the arctan function from Python’s NumPy 1.23 library, as follows:(6)$$\hat{\theta }=\tan^{-1}\left(Output\hat{2}/Output\hat{1}\right)$$
and
(7)$$\hat{P}=\left\{\begin{array}{c} \;\;\left\|\hat{\theta }\right\|/2\pi ,\;\;\;\;\;\;\;\;\;\;\;\;\;\;\;\;\;Output\hat{2}\geq 0\\ \;\;1-\left\|\hat{\theta }\right\|/2\pi ,\;\;Output\hat{2}<0 \end{array}\right..$$

In Equations (6) and (7), $Output\hat{1}$ and $Output\hat{2}$ denote the estimated values of the CSF, and $\hat{P}$ denotes the cGPE value estimated using Bi-LSTM.

As depicted in Figure 4a, when own-subject data are entered into the proposed model, the estimated gait phase values converge well to the ground truth gait. However, when cross-subject data are entered into the proposed model, the error in the estimated gait phase values increases compared to what is shown in Figure 4a, as shown in Figure 4b.

Table 5 and Figure 5 describe the RMSE and R^2^ scores of the cGPE algorithm with respect to the input feature type. However, only the own-subject test results are presented.

First, as depicted in Figure 5, Input #1 exhibits the highest cGPE performance among the five input features, with RMSE and R^2^ scores of 0.794 and 0.906, respectively. In other words, the best cGPE performance is obtained when all FSR data, along with FSR_Fore_, FSR_Back_, and the WC classification results estimated from the previous stage, are used as input data.

The RMSE and R^2^ values for Input #2, which excludes the WC classification results, are 1.015 and 0.880, respectively. In particular, the R^2^ scores for the SA/SD classification results are observed to be 0.9 or less. The RMSE and R^2^ scores for Input #3, which does not include FSR_Fore_ and FSR_Back_, are 1.207 and 0.857, respectively, whereas those for Input #4, which uses only the features of FSR_Fore_ and FSR_Back_, are 1.184 and 0.860, respectively. This indicates that all 14 FSR data points are essential, in addition to the WC classification results, to improve cGPE performance.

Now, we analyze cGPE performance under LW conditions by analyzing the center of pressure (COP) data generated during walking using an AO, as reported in a previous study [32]. As such, cGPE is performed under the conditions of Input #5, which adds COP to Input #1. The RMSE and R^2^ scores are observed to be 1.045 and 0.877, respectively. Compared to Input #1, performance is observed to be degraded—the R^2^ score decreases, and the RMSE increases. This indicates that the COP data do not have a positive impact on the convergence of the CSF value.

Figure 6 depicts the R^2^ scores of the developed algorithm with respect to varying WCs. The average R^2^ score is 0.922 for LW and 0.920 and 0.908 for RA and RD, respectively. Additionally, relatively low average values of 0.889 and 0.892 are observed for SA and SD, respectively. This phenomenon indicates that the cGPE performance is degraded in these two cases because only the fore part of the sole or heel touches the ground while going up or down a staircase, decreasing the fidelity of the aforementioned 14 FSR data.

## 4. Discussion

Yu et al. [28] estimated gait phases for up to 200 ms under the WCs of LW, SA, and SD. Their model comprised one artificial neural network (ANN) for WC classification and another ANN for gait phase estimation. The proposed model used data collected from an IMU affixed to the lower limb joints of subjects. To evaluate the proposed model under each WC, the relative RMSEs (rRMSEs) were calculated, and accuracy was evaluated. The rRMSE was 1.66% for LW, 4.55% for SA, and 3.51% for SD, whereas the R^2^ score was not evaluated. As per the definition by Yu, rRMSE is obtained by dividing the RMSE between the ground truth gait and the cGPE estimated before 200 ms by the average value of the estimated cGPE. In this study, the rRMSE of cGPE was evaluated to be 1.31% for LW, 1.87% for SA, 1.80% for SD, 1.37% for RA, and 1.55% for RD, indicating better performance than the model proposed by Yu. Yu et al. obtained an extremely high WC classification accuracy of 99.55% for standing, level walking, ascending stairs, and descending stairs.

Wu et al. [19] conducted cGPE research under the WCs of LW, SA, and SD. Their proposed model comprised an ANN for WC classification and an AO model for cGPE. The proposed model used hip angle information as input values, and its accuracy was evaluated in terms of the average phase estimation error (APEE), which was calculated based on the error evaluated corresponding to each WC. The calculated APEE values were 6.61% for LW, 4.12% for SA, and 5.42% for SD; however, the R^2^ score was not evaluated. In this study, the APEE of the proposed cGPE algorithm was 0.66% for LW, 0.94% for SA, 0.93% for SD, 1.06% for RA, and 0.78% for RD, indicating lower errors than Wu’s results. In the study by Wu et al., the WC classification accuracy exceeded 92% for standing, level walking, ascending stairs, and descending stairs.

A comparison of the results of the two studies reveals that the cGPE errors reported by Wu et al. are slightly higher than those reported by Yu et al. However, the latter study did not require additional sensors to be attached, unlike the former, in which an IMU was installed to measure joint angle information using an encoder installed on a walking assistance robot.

Choi et al. [27] performed cGPE only under the LW condition using a treadmill. They developed an LSTM-based cGPE algorithm that introduced a domain-adversarial model and conducted a cross-subject test using data obtained from an IMU installed on the femoral region of each subject. Regarding the performance of the proposed algorithm, the R^2^ value was at least 0.916. Choi defined the normalized RMSE (NRMSE) by dividing the RMSE by the difference between the maximum and minimum estimated cGPE values. The NRMSE from the walking experiment performed on a treadmill was used to evaluate the cGPE performance. The NRMSE was observed to be 4.54%. In this study, the NRMSE of the proposed cGPE algorithm is 0.66% for LW, 0.94% for SA, 0.92% for SD, 0.68% for RA, and 0.78% for RD, indicating smaller errors compared to the case of Choi et al. Thus, a capable cGPE algorithm that can be used for cross-subjects has been created.

Meanwhile, Mazon et al. and Feng et al. obtained WC classification accuracies exceeding 95% and 90%, respectively, by classifying WCs using data obtained from an IMU attached to the lower limb [26] and classifying WCs using strain gauges to control prosthetic legs [33].

The proposed cGPE algorithm, developed in this study, meets the performance of previous studies, even in a multi-locomotion task environment based solely on FSR data.

## 5. Conclusions

In this study, a cGPE algorithm is proposed that analyzes FSR data using two Bi-LSTM models. The proposed method represents a single algorithm that applies to paved and unpaved roads and enables GPE under five WCs: LW, SA, SD, RA, and RD. The WC classification accuracy exceeds 90%. The cGPE algorithm, trained on individual subjects, estimates the gait phase with the highest accuracy when it is applied to its own subject.

## Figures and Tables

**Figure 1 sensors-24-06318-f001:**
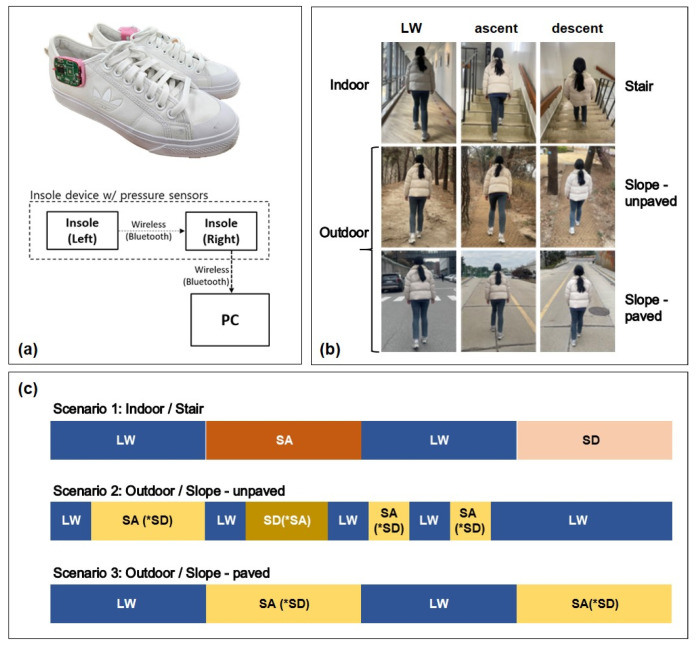
Experimental configuration: (**a**) data acquisition system; (**b**) walking conditions; (**c**) test protocol for the multi-locomotion task. * The reverse direction walking conditions.

**Figure 2 sensors-24-06318-f002:**
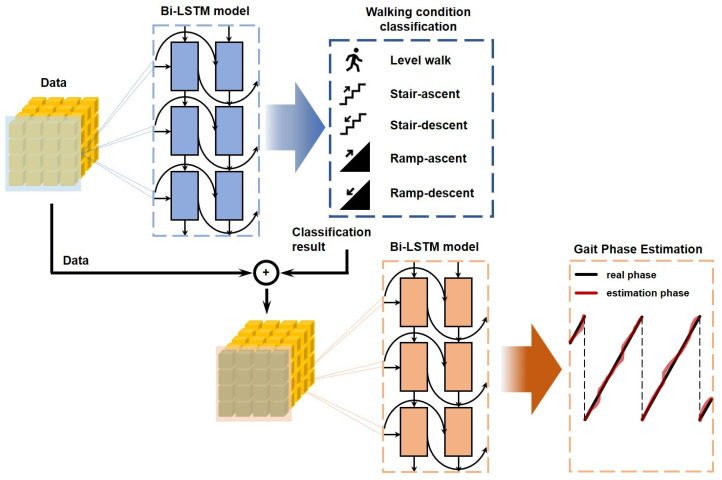
Architecture of the proposed continuous gait phase estimation algorithm using two steps Bi-LSTM.

**Figure 3 sensors-24-06318-f003:**
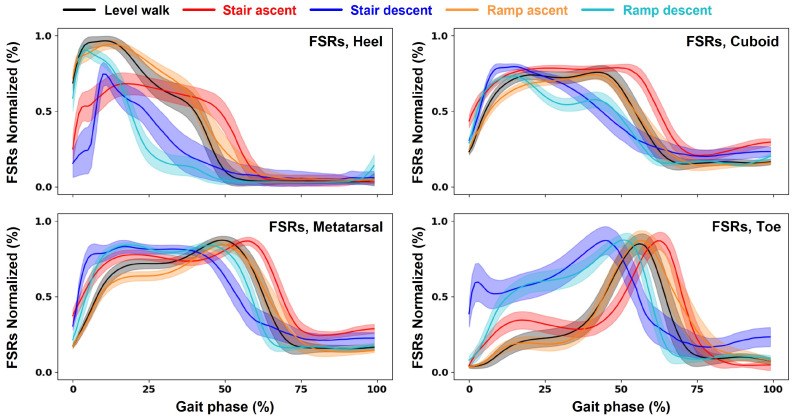
Typical variation of FSR data from Subject 1 measured during multi-locomotion tasks. Data measured at four different locations on the foot while performing walking experiments under five different walking conditions are presented.

**Figure 4 sensors-24-06318-f004:**
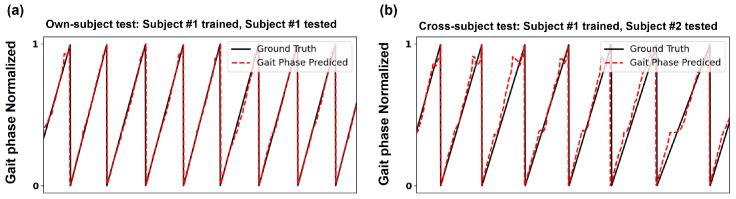
Performance of the proposed cGPE algorithm evaluated using own-subject data and cross-subject data. The own-subject estimation results converge well to the ground truth gait. (**a**) R^2^ score in the own-subject test scenario, where Subject #1 is trained and tested, is 0.895, (**b**) R^2^ score in the cross-subject test scenario, where Subject #1 is trained, and Subject #2 is tested, is 0.830.

**Figure 5 sensors-24-06318-f005:**
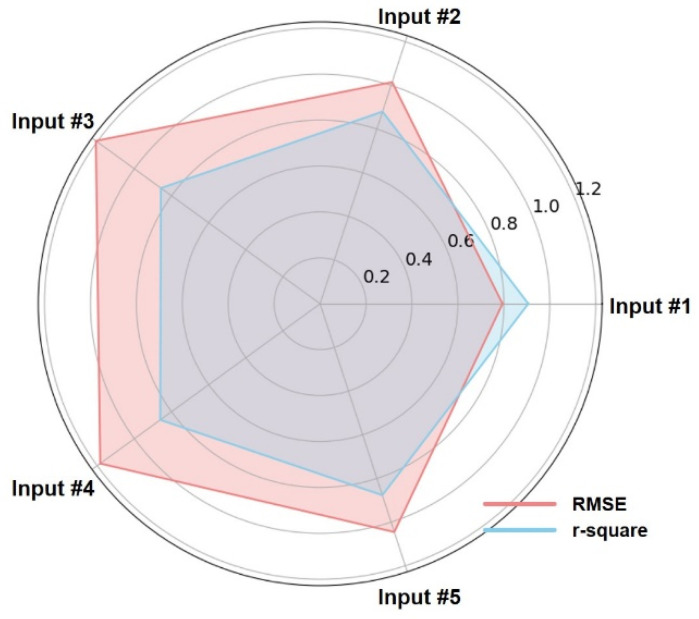
Comparison of RMSE and R^2^ score corresponding to five input features.

**Figure 6 sensors-24-06318-f006:**
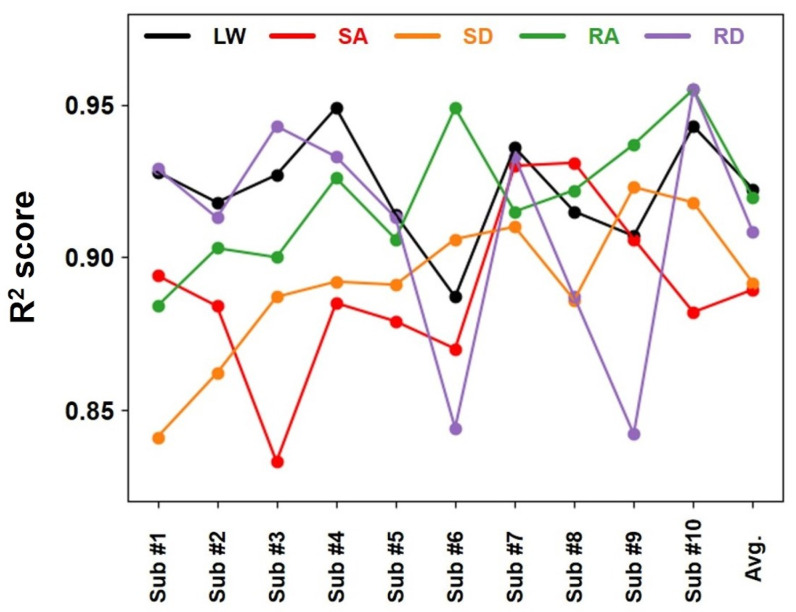
R^2^ scores evaluated in different WCs. Each value is assessed individually for each subject and then averaged.

**Table 1 sensors-24-06318-t001:** Architecture of a Bi-LSTM network for cGPE based on ground reaction force data. Table annotations: 1: sequence size of the classification model; 2: sequence size of the gait phase estimation model; 3: input vector of the classification model (FSR data); 4: input vector of the gait phase estimation model (FSR, FSR_Fore_, FSR_Back_, and classification model’s output results).

Model	Layer	Shape
Walking Condition Classification	Bi-LSTM	(128, 100 (1), 10 (3))
	Drop out	0.25
	Bi-LSTM	64
	Drop out	0.25
	Fully Connected	64
	Output layer	5
Continuous Gait Phase Estimation	Bi-LSTM	(128, 100 (2), 10 (4))
	Drop out	0.25
	Bi-LSTM	32
	Drop out	0.25
	Fully Connected	2
	Output layer	2

**Table 2 sensors-24-06318-t002:** Confusion matrix for walking condition classification.

Actual Class	Predict Class				
	Level Walk	Stair Ascent	Stair Descent	Ramp Ascent	Ramp Descent
Level walk	90 ± 1%	0 ± 0%	0 ± 0%	5 ± 2%	4 ± 2%
Stair ascent	6 ± 2%	91 ± 4%	1 ± 1%	2 ± 2%	1 ± 2%
Stair descent	5 ± 4%	1 ± 1%	90 ± 5%	0 ± 0%	4 ± 4%
Ramp ascent	9 ± 3%	0 ± 0%	0 ± 0%	90 ± 3%	1 ± 1%
Ramp descent	8 ± 3%	0 ± 0%	0 ± 0%	0 ± 0%	91 ± 3%

**Table 3 sensors-24-06318-t003:** RMSE with R^2^ scores in parenthesis of the continuous gait phase estimation algorithm trained on the multi-locomotion task data for Subjects 1–5. Black blocks highlight the own-subject test results.

Subject Used for Learning		Subject Used for Testing									
		#1	#2	#3	#4	#5	#6	#7	#8	#9	#10
#1	LW	0.607 (0.928)	1.750 (0.793)	1.487 (0.824)	2.988 (0.648)	1.083 (0.872)	1.149 (0.864)	0.633 (0.926)	0.824 (0.903)	0.795 (0.907)	0.717 (0.916)
	SA	0.895 (0.894)	1.833 (0.783)	1.277 (0.851)	3.070 (0.638)	2.177 (0.746)	2.438 (0.714)	2.581 (0.695)	2.404 (0.717)	3.936 (0.534)	3.349 (0.604)
	SD	1.359 (0.841)	1.668 (0.804)	1.142 (0.864)	2.968 (0.646)	1.225 (0.855)	1.523 (0.819)	0.956 (0.887)	1.438 (0.830)	1.486 (0.827)	1.523 (0.820)
	RA	0.985 (0.884)	0.669 (0.921)	0.696 (0.918)	2.678 (0.685)	1.134 (0.866)	0.802 (0.906)	0.546 (0.935)	0.379 (0.955)	0.672 (0.921)	0.707 (0.916)
	RD	0.598 (0.929)	1.286 (0.848)	1.464 (0.827)	2.978 (0.648)	1.143 (0.865)	1.503 (0.822)	0.798 (0.906)	1.271 (0.850)	0.988 (0.883)	0.520 (0.938)
#2	LW	0.886 (0.895)	0.692 (0.918)	0.785 (0.907)	1.717 (0.798)	0.819 (0.903)	2.038 (0.759)	1.028 (0.879)	0.850 (0.900)	1.159 (0.864)	2.086 (0.755)
	SA	3.244 (0.617)	0.979 (0.884)	2.549 (0.702)	3.018 (0.644)	2.216 (0.742)	5.199 (0.390)	3.994 (0.529)	3.520 (0.586)	7.061 (0.164)	4.757 (0.437)
	SD	6.430 (0.246)	1.172 (0.862)	1.599 (0.810)	1.977 (0.764)	1.483 (0.825)	1.544 (0.816)	1.667 (0.803)	1.007 (0.881)	3.324 (0.612)	2.117 (0.749)
	RA	1.647 (0.806)	0.821 (0.903)	0.668 (0.921)	1.617 (0.810)	1.311 (0.846)	2.917 (0.658)	1.296 (0.847)	1.272 (0.851)	1.706 (0.800)	1.770 (0.790)
	RD	1.154 (0.864)	0.736 (0.913)	0.757 (0.911)	1.388 (0.836)	0.747 (0.911)	1.745 (0.794)	0.939 (0.889)	1.064 (0.875)	1.231 (0.855)	1.490 (0.824)
#3	LW	1.334 (0.843)	1.129 (0.866)	0.618 (0.927)	5.778 (0.319)	0.793 (0.906)	3.317 (0.608)	1.166 (0.863)	1.150 (0.864)	0.964 (0.887)	1.614 (0.811)
	SA	2.999 (0.646)	2.350 (0.721)	1.424 (0.833)	4.576 (0.461)	1.983 (0.769)	4.529 (0.469)	2.853 (0.663)	2.617 (0.692)	5.447 (0.355)	3.765 (0.555)
	SD	4.245 (0.502)	1.190 (0.860)	0.952 (0.887)	3.593 (0.572)	0.907 (0.893)	2.778 (0.670)	1.518 (0.820)	1.186 (0.860)	2.901 (0.662)	2.703 (0.680)
	RA	2.001 (0.765)	1.546 (0.818)	0.847 (0.900)	3.943 (0.537)	0.931 (0.890)	2.260 (0.735)	1.261 (0.851)	1.422 (0.833)	1.417 (0.834)	1.344 (0.840)
	RD	1.287 (0.848)	1.545 (0.818)	0.558 (0.934)	5.330 (0.369)	0.760 (0.910)	3.262 (0.614)	1.031 (0.878)	1.108 (0.869)	0.981 (0.884)	1.202 (0.858)
#4	LW	1.967 (0.768)	1.507 (0.822)	4.254 (0.498)	0.433 (0.949)	1.991 (0.765)	3.545 (0.581)	1.012 (0.881)	0.659 (0.922)	0.933 (0.890)	0.991 (0.884)
	SA	1.892 (0.777)	2.989 (0.646)	1.800 (0.790)	0.977 (0.885)	1.950 (0.773)	2.757 (0.677)	2.557 (0.698)	1.122 (0.868)	4.581 (0.457)	4.231 (0.500)
	SD	11.929 (−0.399)	1.679 (0.803)	2.754 (0.673)	0.906 (0.892)	3.081 (0.636)	1.623 (0.807)	1.143 (0.865)	1.185 (0.860)	1.760 (0.795)	1.591 (0.812)
	RA	2.782 (0.673)	0.936 (0.890)	1.177 (0.861)	0.628 (0.926)	2.050 (0.759)	2.748 (0.677)	0.894 (0.894)	0.537 (0.937)	1.201 (0.859)	0.814 (0.903)
	RD	2.319 (0.726)	1.189 (0.860)	5.750 (0.322)	0.564 (0.933)	4.150 (0.508)	2.594 (0.693)	0.912 (0.892)	1.231 (0.855)	1.037 (0.877)	0.731 (0.914)
#5	LW	1.155 (0.864)	1.013 (0.880)	0.970 (0.886)	1.654 (0.805)	0.727 (0.914)	3.501 (0.587)	1.309 (0.846)	1.531 (0.819)	1.215 (0.857)	2.081 (0.756)
	SA	2.745 (0.676)	1.316 (0.844)	1.389 (0.838)	2.289 (0.730)	1.040 (0.879)	4.116 (0.517)	1.964 (0.768)	1.586 (0.813)	5.551 (0.343)	4.166 (0.507)
	SD	9.937 (−0.165)	1.268 (0.851)	1.712 (0.797)	1.921 (0.771)	0.919 (0.891)	2.451 (0.709)	1.441 (0.829)	1.449 (0.829)	2.686 (0.687)	2.620 (0.690)
	RA	1.598 (0.812)	0.891 (0.895)	0.643 (0.924)	1.260 (0.852)	0.798 (0.906)	2.206 (0.741)	1.223 (0.855)	1.506 (0.823)	1.889 (0.778)	1.388 (0.835)
	RD	1.532 (0.819)	0.815 (0.904)	1.031 (0.878)	1.191 (0.859)	0.736 (0.913)	3.445 (0.593)	1.166 (0.862)	1.811 (0.787)	1.290 (0.848)	1.234 (0.854)

**Table 4 sensors-24-06318-t004:** RMSE values with R^2^ scores in parentheses of the continuous gait phase estimation algorithm trained on the multi-locomotion task data for Subjects 6–10. Black blocks highlight the own-subject test results.

Subject Used for Learning		Subject Used for Testing									
		#1	#2	#3	#4	#5	#6	#7	#8	#9	#10
#6	LW	2.010 (0.763)	2.631 (0.689)	1.888 (0.777)	2.233 (0.737)	2.504 (0.705)	0.955 (0.887)	1.946 (0.771)	0.795 (0.906)	1.242 (0.854)	0.587 (0.931)
	SA	10.036 (−0.185)	3.164 (0.625)	4.864 (0.431)	4.742 (0.441)	3.057 (0.644)	1.104 (0.870)	3.050 (0.640)	4.529 (0.467)	2.269 (0.731)	4.609 (0.455)
	SD	12.719 (−0.491)	3.866 (0.546)	2.267 (0.731)	1.925 (0.770)	7.350 (0.132)	0.792 (0.906)	1.325 (0.843)	1.496 (0.823)	1.344 (0.843)	1.350 (0.840)
	RA	10.081 (−0.186)	5.471 (0.355)	2.817 (0.668)	2.471 (0.710)	4.541 (0.465)	0.438 (0.949)	2.642 (0.688)	0.505 (0.941)	0.737 (0.913)	0.642 (0.924)
	RD	1.648 (0.806)	2.321 (0.726)	1.083 (0.872)	1.822 (0.784)	4.646 (0.449)	1.320 (0.844)	2.885 (0.659)	1.087 (0.872)	1.395 (0.835)	0.647 (0.923)
#7	LW	1.985 (0.766)	1.435 (0.828)	2.042 (0.759)	5.838 (0.312)	1.442 (0.930)	4.309 (0.491)	0.545 (0.936)	0.654 (0.923)	0.745 (0.913)	1.148 (0.865)
	SA	7.603 (0.103)	3.618 (0.571)	1.728 (0.798)	6.780 (0.201)	4.002 (0.534)	4.291 (0.497)	0.592 (0.930)	2.222 (0.739)	3.723 (0.559)	2.398 (0.716)
	SD	9.771 (−0.146)	2.610 (0.693)	1.188 (0.859)	5.720 (0.318)	2.394 (0.717)	4.435 (0.473)	0.756 (0.910)	1.260 (0.851)	2.103 (0.755)	1.791 (0.788)
	RA	2.753 (0.676)	1.436 (0.831)	1.636 (0.807)	4.516 (0.469)	2.105 (0.752)	2.654 (0.688)	0.722 (0.915)	0.903 (0.894)	1.211 (0.858)	1.120 (0.867)
	RD	2.248 (0.735)	1.592 (0.812)	2.455 (0.710)	5.248 (0.379)	2.032 (0.759)	4.332 (0.488)	0.564 (0.933)	1.492 (0.824)	0.949 (0.888)	0.811 (0.904)
#8	LW	2.270 (0.732)	1.476 (0.825)	2.771 (0.673)	5.000 (0.411)	1.303 (0.846)	2.434 (0.713)	1.269 (0.851)	0.721 (0.915)	1.662 (0.805)	1.252 (0.853)
	SA	5.530 (0.347)	4.611 (0.453)	2.857 (0.666)	4.400 (0.482)	2.248 (0.738)	4.138 (0.515)	1.113 (0.869)	0.584 (0.931)	3.685 (0.564)	2.289 (0.729)
	SD	4.261 (0.500)	1.829 (0.785)	2.621 (0.689)	4.751 (0.433)	3.654 (0.568)	3.951 (0.530)	1.059 (0.875)	0.962 (0.886)	3.144 (0.633)	2.277 (0.730)
	RA	2.255 (0.735)	1.541 (0.818)	1.947 (0.771)	4.832 (0.432)	2.418 (0.715)	7.870 (0.076)	1.485 (0.824)	0.667 (0.922)	2.498 (0.707)	0.997 (0.882)
	RD	2.280 (0.731)	2.070 (0.756)	3.704 (0.563)	4.161 (0.508)	1.209 (0.857)	2.581 (0.695)	1.121 (0.868)	0.957 (0.887)	1.452 (0.828)	1.044 (0.876)
#9	LW	2.488 (0.707)	2.873 (0.660)	3.336 (0.606)	1.984 (0.766)	4.820 (0.432)	1.703 (0.799)	0.909 (0.893)	0.705 (0.917)	0.794 (0.907)	0.420 (0.951)
	SA	4.358 (0.486)	2.717 (0.678)	2.606 (0.695)	1.849 (0.782)	6.069 (0.293)	2.174 (0.745)	1.508 (0.822)	1.796 (0.789)	0.797 (0.906)	3.075 (0.636)
	SD	12.343 (−0.447)	3.041 (0.643)	1.454 (0.827)	1.593 (0.810)	2.297 (0.729)	1.136 (0.865)	1.457 (0.827)	1.144 (0.865)	0.660 (0.923)	1.765 (0.791)
	RA	3.169 (0.627)	4.738 (0.442)	2.714 (0.680)	4.092 (0.519)	2.845 (0.665)	6.273 (0.264)	2.096 (0.752)	0.406 (0.952)	0.535 (0.937)	0.422 (0.950)
	RD	2.399 (0.717)	3.354 (0.605)	1.859 (0.781)	2.477 (0.707)	7.358 (0.128)	1.654 (0.804)	0.851 (0.900)	1.176 (0.861)	1.341 (0.842)	0.456 (0.946)
#10	LW	2.032 (0.760)	1.821 (0.785)	3.630 (0.572)	4.447 (0.476)	3.589 (0.577)	3.137 (0.630)	0.518 (0.939)	0.635 (0.925)	0.837 (0.902)	0.486 (0.943)
	SA	3.742 (0.558)	3.304 (0.608)	2.832 (0.669)	4.341 (0.489)	2.243 (0.739)	5.493 (0.355)	1.166 (0.862)	1.722 (0.797)	5.566 (0.341)	0.995 (0.882)
	SD	8.383 (0.017)	1.896 (0.608)	1.969 (0.766)	3.500 (0.583)	3.579 (0.577)	5.342 (0.365)	0.736 (0.913)	1.077 (0.873)	2.249 (0.738)	0.689 (0.918)
	RA	8.031 (0.055)	1.508 (0.822)	3.537 (0.583)	2.617 (0.693)	8.233 (0.030)	1.933 (0.773)	0.394 (0.953)	0.667 (0.922)	0.829 (0.903)	0.382 (0.955)
	RD	2.339 (0.724)	2.289 (0.730)	2.935 (0.654)	3.872 (0.542)	6.821 (0.192)	4.114 (0.514)	0.785 (0.907)	1.089 (0.872)	1.085 (0.872)	0.377 (0.955)

**Table 5 sensors-24-06318-t005:** RMSE values with R^2^ scores in parentheses of the continuous gait phase estimation algorithm. Each model is trained on five types of input features.

Input Data Used for Learning		Subject Used for Own-Subject Prediction									
		#1	#2	#3	#4	#5	#6	#7	#8	#9	#10
each FSR FSR_Fore_ FSR_Back_ WC (input#1)	LW	0.607 (0.928)	0.692 (0.918)	0.618 (0.927)	0.433 (0.949)	0.727 (0.914)	0.955 (0.887)	0.545 (0.936)	0.721 (0.915)	0.794 (0.907)	0.486 (0.943)
	SA	0.895 (0.894)	0.979 (0.884)	1.424 (0.833)	0.977 (0.885)	1.040 (0.879)	1.104 (0.870)	0.592 (0.930)	0.584 (0.931)	0.797 (0.906)	0.995 (0.882)
	SD	1.359 (0.841)	1.172 (0.862)	0.952 (0.887)	0.906 (0.892)	0.919 (0.891)	0.792 (0.906)	0.756 (0.910)	0.962 (0.886)	0.660 (0.923)	0.689 (0.918)
	RA	0.985 (0.884)	0.821 (0.903)	0.847 (0.900)	0.628 (0.926)	0.798 (0.906)	0.438 (0.949)	0.722 (0.915)	0.667 (0.922)	0.535 (0.937)	0.382 (0.955)
	RD	0.598 (0.929)	0.736 (0.913)	0.558 (0.934)	0.564 (0.933)	0.736 (0.913)	1.320 (0.844)	0.564 (0.933)	0.957 (0.887)	1.341 (0.842)	0.377 (0.955)
each FSR FSR_Fore_ FSR_Back_ (input#2)	LW	0.799 (0.906)	0.827 (0.927)	0.632 (0.905)	0.808 (0.905)	1.158 (0.863)	0.886 (0.895)	0.645 (0.924)	0.835 (0.902)	0.762 (0.911)	0.372 (0.956)
	SA	1.366 (0.839)	2.786 (0.670)	1.403 (0.797)	1.822 (0.785)	1.887 (0.780)	1.997 (0.766)	0.658 (0.922)	0.951 (0.888)	0.945 (0.888)	1.478 (0.825)
	SD	1.661 (0.805)	0.936 (0.890)	1.060 (0.879)	1.311 (0.844)	1.541 (0.818)	0.821 (0.902)	0.662 (0.922)	0.824 (0.903)	0.673 (0.922)	0.933 (0.889)
	RA	1.269 (0.851)	1.367 (0.839)	0.958 (0.940)	0.842 (0.901)	1.226 (0.856)	0.474 (0.944)	0.723 (0.914)	0.764 (0.910)	0.670 (0.921)	0.445 (0.947)
	RD	0.754 (0.911)	1.059 (0.875)	0.548 (0.888)	0.664 (0.921)	1.082 (0.872)	1.424 (0.832)	0.590 (0.930)	0.815 (0.904)	1.206 (0.857)	0.424 (0.950)
each FSR WC (input#3)	LW	1.001 (0.882)	1.254 (0.852)	1.277 (0.849)	0.849 (0.900)	1.003 (0.882)	0.940 (0.889)	1.077 (0.873)	1.580 (0.814)	0.596 (0.930)	1.023 (0.880)
	SA	1.814 (0.709)	1.316 (0.844)	1.630 (0.809)	1.733 (0.796)	1.686 (0.804)	1.249 (0.853)	1.390 (0.836)	1.186 (0.861)	1.004 (0.881)	1.279 (0.849)
	SD	2.480 (0.709)	1.240 (0.854)	1.484 (0.824)	0.985 (0.883)	1.357 (0.840)	0.966 (0.885)	1.392 (0.835)	1.142 (0.865)	1.042 (0.878)	1.126 (0.867)
	RA	1.525 (0.820)	1.589 (0.813)	1.333 (0.843)	0.991 (0.884)	0.944 (0.889)	0.783 (0.908)	0.995 (0.882)	1.385 (0.837)	0.852 (0.932)	0.995 (0.882)
	RD	1.123 (0.867)	1.237 (0.854)	1.137 (0.866)	0.842 (0.900)	1.123 (0.867)	1.430 (0.831)	0.886 (0.895)	1.471 (0.827)	0.781 (0.908)	0.827 (0.902)
FSR_Fore_ FSR_Back_ WC (input#4)	LW	0.786 (0.907)	1.053 (0.875)	1.133 (0.866)	1.265 (0.851)	1.468 (0.827)	0.966 (0.886)	0.844 (0.901)	1.570 (0.815)	0.593 (0.930)	0.930 (0.891)
	SA	1.346 (0.841)	1.515 (0.820)	1.647 (0.807)	2.183 (0.743)	2.041 (0.762)	1.410 (0.835)	0.806 (0.905)	1.762 (0.793)	0.905 (0.893)	1.260 (0.851)
	SD	1.366 (0.840)	1.113 (0.869)	0.988 (0.883)	0.957 (0.886)	1.635 (0.807)	0.887 (0.895)	0.913 (0.892)	1.526 (0.820)	1.052 (0.877)	0.916 (0.891)
	RA	1.210 (0.858)	1.779 (0.790)	1.570 (0.815)	1.388 (0.837)	1.441 (0.830)	0.719 (0.916)	0.817 (0.903)	1.514 (0.822)	0.586 (0.931)	0.813 (0.903)
	RD	0.715 (0.916)	1.253 (0.852)	0.909 (0.893)	1.003 (0.881)	1.584 (0.812)	1.403 (0.834)	0.768 (0.909)	1.490 (0.824)	0.783 (0.908)	0.608 (0.928)
FSR_Fore_ FSR_Back_ COP WC (input#5)	LW	0.870 (0.897)	1.048 (0.876)	0.632 (0.925)	0.420 (0.951)	1.581 (0.814)	1.093 (0.871)	0.806 (0.905)	1.034 (0.878)	1.139 (0.866)	0.395 (0.954)
	SA	2.190 (0.741)	1.336 (0.842)	1.403 (0.836)	1.289 (0.848)	2.006 (0.766)	1.176 (0.862)	0.763 (0.910)	1.308 (0.846)	0.929 (0.890)	0.883 (0.896)
	SD	1.554 (0.818)	1.198 (0.859)	1.060 (0.874)	0.778 (0.907)	1.518 (0.821)	0.640 (0.924)	0.791 (0.906)	0.844 (0.900)	1.085 (0.874)	0.806 (0.905)
	RA	0.715 (0.916)	1.215 (0.857)	0.958 (0.887)	0.554 (0.935)	1.850 (0.782)	0.373 (0.956)	0.963 (0.886)	1.071 (0.874)	0.792 (0.907)	0.444 (0.947)
	RD	0.836 (0.901)	1.056 (0.876)	0.548 (0.935)	0.702 (0.917)	1.661 (0.803)	1.540 (0.818)	0.811 (0.904)	0.923 (0.891)	2.271 (0.732)	0.408 (0.952)

## Data Availability

The data presented in this study are available upon request from the corresponding author. The data are not publicly available, owing to ethical concerns, as they were obtained in a clinical trial.

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
