# Peer review of "Continuous Gait Phase Estimation for Multi-Locomotion Tasks Using Ground Reaction Force Data"

_sensors, 2024, doi:10.3390/s24196318_

Round 1

Reviewer 1 Report

Comments and Suggestions for Authors

This paper demonstrates a certain level of innovation and presents a comprehensive theoretical framework, contributing new insights to the field. However, there are two main issues that need to be addressed. First, the labels (a), (b), and (c) in Figure 2 are not clearly marked, which may cause confusion for readers when interpreting the figure. It is recommended to add clear labels to enhance the readability and clarity of the figure. Second, the description of the data collection process lacks a clear indication of the specific areas on the foot sole where data needs to be collected. To better illustrate the critical areas for data collection, it would be beneficial to include a diagram that clearly shows the specific locations and regions on the foot sole. These improvements would enhance the overall clarity and academic rigor of the paper.

Comments on the Quality of English Language

The English language used in the paper is generally clear and fluent, with appropriate grammar and vocabulary that effectively convey the research content and key ideas. The structure of the paper is well-organized, with clear logic and accurate use of terminology, demonstrating the author’s good command of professional English. Although there are a few expressions that could be further refined, the overall language quality is high and does not hinder the reader’s understanding and evaluation of the research work. It is suggested to make minor adjustments in the final version to further enhance the precision and fluency of the expression.

Author Response

Please see the attached Q&A document.

Reviewer 2 Report

Comments and Suggestions for Authors

Overview:

- The current study is regarding estimating gait phase for a variety of tasks using data from force sensing resistors in a shoe insole to develop a BiLSTM algorithm. This is an important area of research, as measuring research participants outside of the laboratory environments is a critical area of interest and important to develop sound methods to do this accurately. Overall, the study is well designed and executed. However, there are a number of methodological details around the study before it is ready for publication.

- I think the biggest issue I have with the paper in its current form is I don’t think it is appropriate to call the data that comes from the FSRs in the insoles “ground reaction force” as it is not measured at the ground (via mounted force platforms). Previous studies that have studied the relationship between in-lab force platforms and insole systems have found relatively high error (on the order of ~15-20%). They are related and generally should be pretty correlated, but ultimately I don’t think it should be presented as using GRFs. There are other studies that have utilized deep learning techniques to create models with IMU data to get estimates of GRF and initially I thought your study would be doing something similar. So I think it would be best to clarify this in some way that it an estimated ground reaction force from the FSR. Or, alternatively, if you have any data or another reference comparing to a ground truth force platform that shows validity between systems, that also would help resolve this issue.

Introduction:

- I think you could combine paragraphs as several are single sentences (e.g. 24-28, 63-72 should be combined into single paragraphs, respectively). This would make things a bit easier to read.

- Line 72: “summarized as follows:”

Methods:

- Would incline and decline be more appropriate terminology than ramp ascent and descent?

- I know you reference the previous study [32] but it would be nice to have any additional information as to sensitivity of the underlying sensors relating to resolution of the data, minimum/maximum detectable forces, etc. And as previously noted, if you have any in-lab validation for this device, providing that would be helpful.

- Is 100 Hz high enough of a sampling rate? For doing gait phase detection, likely enough for walking, but assuming the insoles provide a valid measure of GRF and you wanted to use these in some kind of clinical population to measure and analyze GRFs longitudinally, 100 Hz may not provide the sensitivity needed to get accurate measures.

- Is the walking speed the same (or assumed to be the same) for each of the tasks? I realize it’s not a main objective of this study but were speeds relatively stable across each of the 3 trials for each task?   

- For stair ascent/descent, was it one long continuous stairway or were there landings? How many floors and/or continuous stairs? I think this is an important thing to report but will revisit why I’m asking this in results.

- Line 138: I’m not sure that application is the correct word to use here.

- Details on how the model were trained seems sparse. Learning rate?

Results:

- Looking at the confusion matrix, the stair accent loses some accuracy to level walking, which is why I asked if the stairs used had a landing/how long they are. Essentially for that condition is all up and down or is there a flat part in the middle that could be adversely affecting the accuracy here.

- I don’t think I follow why the subjects are split into two groups for cross validation. Why not use all 10?

- Line 246: I’m not sure I follow what you’re saying here.

- I think Figure 6 would be better presented as a table.

Discussion:

- The discussion overall is a bit hard to follow. I understand you are trying to compare these studies against the current study but how well the current study compares is not completely clear. It appears that the current study meets or exceeds previous studies, but I think this section could be rewritten for clarity.

- What’s the advantage of using just insoles over a sensor-based system? I think that’s not entirely established. I understand the limitations of the sensor systems may not completely account for inclines, stairs, etc (hence the current study) but what is the value-added of insoles versus a sensor?

- Are there any limitations to note?

- I think the conclusions section should simply be rewritten to be a single paragraph.

Comments on the Quality of English Language

Generally fine. Nothing major to note, mostly just condensing some of the single sentence paragraphs together as appropriate. 

Author Response

Please see the attached Q&A document.

Reviewer 3 Report

Comments and Suggestions for Authors

Detailed comments see attachments

Comments on the Quality of English Language

The English language can be improved further.

Author Response

Please see the attached Q&A document.

Round 2

Reviewer 2 Report

Comments and Suggestions for Authors

Updated article has addressed my comments and concerns adequately. 

Reviewer 3 Report

Comments and Suggestions for Authors

The current form can be accepted